# Ecological suitability evaluation of traditional village locations in Jiangxi Province based on multi-model integration using artificial intelligence

Cheng Zhang[1,2]*, Jinlin Teng[1,2], Peilin Liu[1,2], Chunqing Liu [1,2]

1 College of Landscape Architecture and Art, Jiangxi Agricultural University, Nanchang, China, 2 Jiangxi Rural Culture Development Research Center, Nanchang, China

* liuchunqing@jxau.edu.cn

**Data availability statement:** The dataset is available for download for free use at this link: https://github.com/Jack13026212687/ Zhang-C-Paper-data.git.

## Abstract

Traditional villages have evolved over time to adapt to their environmental characteristics, demonstrating high ecological suitability. Ideal village locations not only provide comfortable living spaces but also ensure safety and sustainability, reflecting the ancestors' profound understanding of the natural environment and ecological wisdom. This study employs an artificial intelligence-based multi-model integration approach to evaluate the ecological suitability of 413 traditional village sites in Jiangxi Province. Key influencing factors are identified, and the ecological wisdom of ancestral site selection is analyzed, resulting in an ecological suitability evaluation map for traditional village locations in Jiangxi Province. The study draws upon environmental characteristic data of Jiangxi Province, including topography, climate, habitat quality, land use, air quality, vegetation cover, and river network density. GIS technology is utilized for spatial analysis and result visualization, with raster data being extracted and standardized. Machine learning methods, such as Random Forest, Support Vector Machine, and Gradient Boosting Decision Trees, along with deep learning methods like Convolutional Neural Networks and Multi-layer Perceptrons, are applied. Multi-model integration techniques combine diverse predictive outputs, thereby enhancing the overall accuracy and robustness of ecological suitability evaluations. Experimental results indicate that elevation, slope, habitat quality, actual distance to water bodies, and average temperature are the main influencing factors for village site selection. The multi-model integration method performs excellently in evaluating ecological suitability, effectively identifying key ecological factors. The model's accuracy and reliability are verified through confusion matrix, feature importance analysis, and ROC curve. By analyzing the impact weights of various ecological factors, this study constructs a Composite Suitability Index (CSI) and generates an ecological suitability evaluation map that clearly displays suitability levels. This provides a scientific basis for the protection and rational development of traditional villages and serves as a reference for ecological site selection studies in other regions.

**Funding:** We thank the National Natural Science Foundation of China for supporting this research through the projects "Gene Identification and Map Construction of Traditional Rural Settlement Landscapes in the Ganjiang River Basin" (Serial No. 51968026) and "Research on the Visual Perception, Quantitative Characterization, and Visual Evaluation of Traditional Village Landscape Resources in Ganjiang River Basin" (Serial No. 52268012). We also acknowledge the support of Jiangxi Rural Culture Development Research Center. We appreciate the technical assistance provided by the GIS and Remote Sensing Laboratory at Jiangxi Agricultural University. Special thanks to all members of the research team for their valuable discussions and contributions to the project.

**Competing interests:** No authors have competing interests.

## Introduction

Traditional villages in China are the product of thousands of years of agrarian civilization, embodying rich historical, cultural, and ecological wisdom, and illustrating the concept of harmonious coexistence between humans and nature. This ecological wisdom refers to adaptive knowledge formed through generations of interaction with the environment, including site practices such as building on higher ground to avoid floods, orienting dwellings for optimal sunlight and ventilation, and preserving vegetation buffers. These vernacular strategies, refined over time, reflect both experiential learning and a scientifically grounded response to local ecological conditions [1,2]. In recent years, as urbanization has accelerated, traditional villages face dual challenges: ecological degradation and cultural heritage loss, making ecological suitability evaluation an increasingly critical focus in academic research and conservation practices [3,4]. Scholars, both domestically and internationally, have explored the site selection of traditional villages using various approaches, ranging from empirical observation to data-driven methods, examining factors related to environmental, historical, and cultural landscapes [5,6]. For instance, in 2019, Zeng Huizi et al. studied traditional villages in the Yuan River Basin of Hunan Province and found that their site selection favored "living by water" [7]. In 2020, Ma Yu et al. employed GIS technology to analyze the relationship between village sites and natural factors in Ningai Village, Shanxi Province [8]. That same year, another study introduced a machine learning approach based on the Gaussian Mixture Model (GMM) to better understand the inherent logic behind the site selection of historical rural settlements [9]. In 2021, Zhao Ke et al. proposed a new method for evaluating land ecological suitability to address the subjectivity involved in factor selection and weight assignment [10]. In 2022, Haoran used spatial analysis to explore the distribution characteristics of traditional villages in China, identifying annual precipitation, average annual temperature, and river density as key factors influencing village distribution [11]. In 2023, Yang et al. examined the site selection of villages along the Great Wall, using a binary logistic regression model to analyze factors such as elevation, slope, and aspect [12,13]. Although these studies address various aspects, they generally lack a systematic analysis of the combined effects of multiple factors.With advances in computer technology and artificial intelligence, machine learning and deep learning methods have introduced new perspectives and approaches to the study of traditional villages [14–18]. Rational site selection based on ecological suitability can not only help protect the cultural heritage of traditional villages but also optimize resource allocation and improve residents' quality of life [19]. Furthermore, national conservation policies, such as the "Guiding Opinions of the State Council on Strengthening the Protection of Traditional Villages" issued in 2012, have further advanced research on the ecological suitability evaluation of traditional villages [20]. Scientifically assessing the ecological suitability of traditional villages provides a reference for policy-making and offers scientific guidance for their protection and sustainable development.

Existing studies on traditional village site selection generally follow one of four methodological paths: empirical and historical analyses that rely on field surveys or archival records; rule-based evaluation methods such as expert scoring systems, which are often constrained by subjectivity; GIS-based overlay techniques that integrate spatial datasets but typically assume fixed weights and linear relationships; and machine learning or deep learning models that capture nonlinear interactions and enhance predictive accuracy. While each method offers useful insights, their isolated application limits the capacity to capture the complex ecological adaptability inherent in traditional village landscapes.While existing studies have made progress in analyzing ecological factors in the site selection of traditional villages, most are

limited to single factors or traditional GIS-based spatial analysis methods, making it difficult to reveal the synergistic effects of multiple factors [21]. These methods face limitations in handling data complexity and high-dimensional spatial data, which constrains the accuracy of suitability evaluations [22,23]. Moreover, current research lacks an in-depth exploration of ecological adaptability patterns, making it challenging to meet the practical needs of conserving traditional village ecosystems. This study employs an artificial intelligence-based multi-model integration approach combined with GIS technology to systematically evaluate the ecological suitability of site selection for 413 traditional villages in Jiangxi Province. The main objectives are to identify key influencing factors, analyze the ecological wisdom embedded in the site selection of traditional villages, and clarify the underlying patterns and causes. Specifically, this study uses multi-model integration and GIS to analyze the impact of various ecological factors on village site selection, providing a scientific basis for the conservation and planning of traditional villages and further exploring the ecological wisdom reflected in the site selection process.

This study addresses the following research questions: 1) What are the key ecological factors influencing the site selection of traditional villages in Jiangxi Province? 2) How effective is the multi-model integration approach in evaluating ecological suitability? 3) What ecological wisdom is reflected in the ancestral site selection of traditional villages? This research aims to provide scientific guidance for the conservation and sustainable development of traditional villages and to offer a reference framework for ecological site selection studies in other regions.

## Study area overview

Jiangxi Province is located in southeastern China, on the southern bank of the middle and lower reaches of the Yangtze River, covering a total area of approximately 166,900 square kilometers. Historically, it has been known as "the head of Wu and the tail of Chu, the gate of Yue and the court of Min." Jiangxi is bordered by Zhejiang and Fujian to the east, Guangdong to the south, Hunan to the west, and Hubei and Anhui to the north. The provincial capital, Nanchang, serves as the political, economic, and cultural center of the province. The terrain of Jiangxi gradually decreases from south to north, presenting an "east high, west low" topography, with mountains and hills accounting for 69.6% of the total area. Major mountain ranges include the Wuyi Mountains, Luoxiao Mountains, and Jiuling Mountains. The main river is the Gan River, which runs through the entire province, and Poyang Lake, the largest freshwater lake in China, is also located here. Jiangxi has a subtropical monsoon climate with distinct seasons, an average annual temperature of 16–18 °C, and an average annual precipitation of 1,400–1,900 mm. The province boasts high vegetation cover and abundant forest resources, making it an important forestry province in China. Traditional villages, as significant cultural heritage sites in Jiangxi Province, preserve a wealth of historical culture and ecological wisdom. As of 2024, there are 413 traditional villages in Jiangxi Province, distributed across 11 prefecture-level cities, mainly concentrated in the hilly and mountainous areas of southern, central, and northeastern Jiangxi (Fig 1).

## Data sources and research methods

### Data sources

This study incorporates multi-source heterogeneous geospatial data, including remote sensing, meteorological, and fundamental geographic data. The data types encompass Points of Interest (POI), Digital Elevation Model (DEM), population statistics, remote sensing

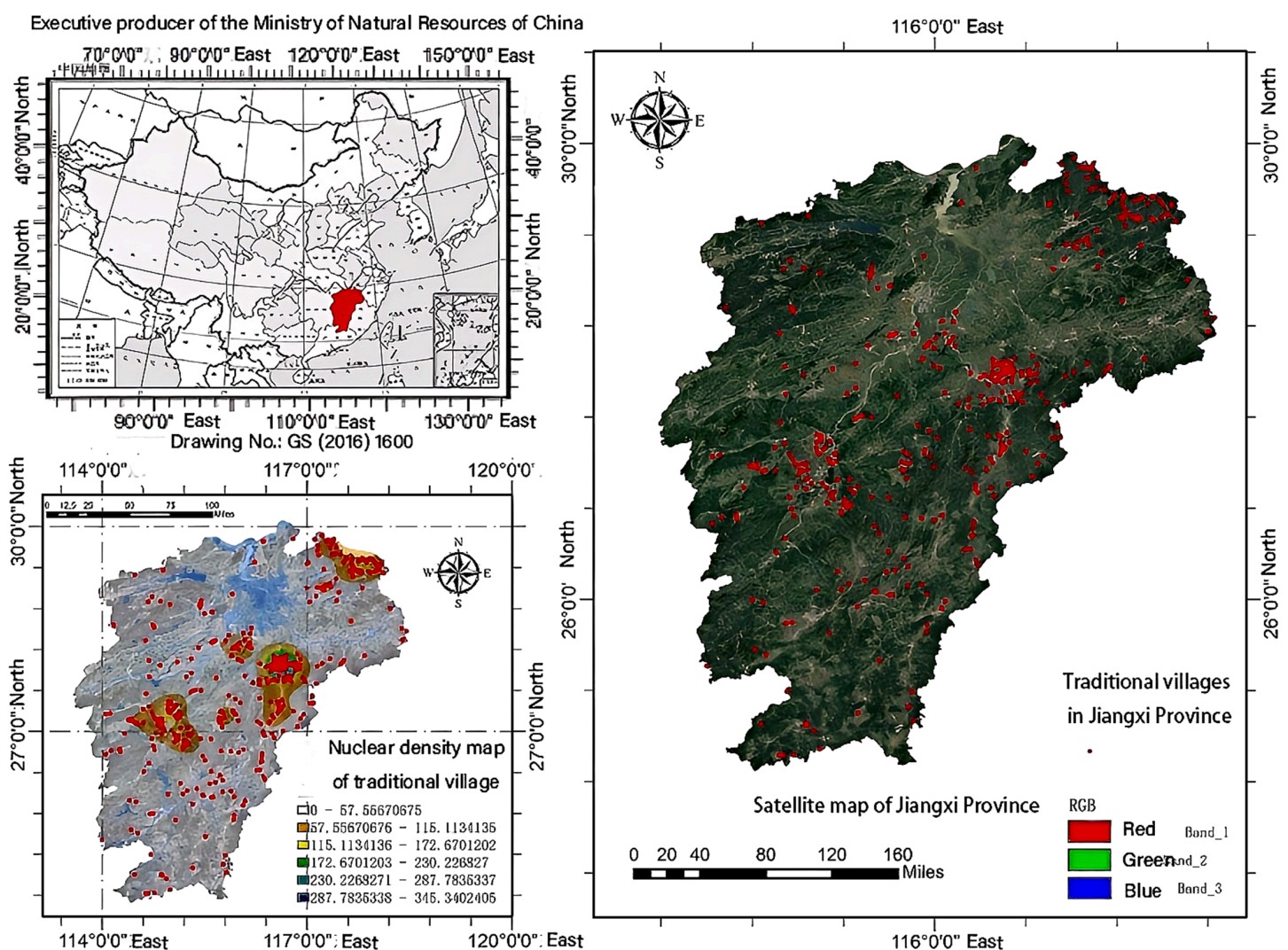

**Fig 1. Site distribution map and nuclear density analysis map of traditional villages in Jiangxi Province, ChinaSite distribution map and nuclear density analysis map of traditional villages in Jiangxi Province, China.** Map data is sourced from Landsat 8 satellite imagery (year: 2021) and overlaid with administrative boundaries obtained from China's National Geomatics Center (2021).

imagery, road networks, river systems, land use, meteorological data, administrative divisions, and ecosystem service values. Data sources are detailed in Table 1. Although the majority of datasets used in this study are from 2020 to 2022, long-term ecological trends in Jiangxi Province confirm their representativeness. As shown in Fig 2, remote sensing and meteorological records from 1995 to 2025 reveal that NDVI values, annual average temperature, and precipitation levels in traditional village areas have remained within a narrow and stable range. This indicates minimal environmental fluctuation in vegetation productivity and climate conditions over the past three decades. Historical reconstructions further show that temperature anomalies in eastern China have remained within ±1°C over the last two millennia [24–26], that land cover and ecosystem service values in Jiangxi have changed gradually without abrupt shifts [27,28], and that the hydroclimatic system in the Ganjiang River Basin

**Table 1. Sources of research data.**

| Data Type | Data Year | Data Source | Data Description |
|---|---|---|---|
| Meteorological Data | 2022 | China Meteorological Annual Spatial Interpolation Dataset (http://www.resdc.cn/DOI) | Represents regional climate conditions, including annual average temperature, precipitation, wind speed, and humidity. IDW interpolation using 5 stations, downscaling with delta algorithm, final raster resolution 0.1°*0.1°. |
| Geological Hazard Data | 2019 | Resource and Environment Data Cloud Platform, CAS (http://www.resdc.cn/) | Data collected up to 2019, covering geological hazards such as collapses, ground fissures, subsidence, sinkholes, landslides, and debris flows. |
| Remote Sensing Data | 2022 | Geospatial Data Cloud (http://www.gscloud.cn/) | 30m Landsat 8 OLI_TIRS data used to calculate the Normalized Difference Vegetation Index (NDVI). |
| Land Cover Data | 2022 | National Tibetan Plateau Data Center (https://data.tpdc.ac.cn/ login) | 30m Landsat 8 OLI_TIRS data used to calculate the Normalized Difference Vegetation Index (NDVI). |
| Ecosystem Service Value Data | 2020 | Resource and Environment Science Data Center, CAS (https://www.resde.cn) | Reflects regional ecosystem service functions and values, representing ecological resource endowment. |
| DEM Data | 2022 | Microtopography 4.3 | ASTER GDEM V3 30m spatial resolution, reflects regional elevation information. |
| Basic vector data | 2022 | Microtopography 4.3 | Obtained hydrological and road network data, calculated water system density, unified coordinate system: WGS1984. |
| Land Use Data | 2020 | Land Use Remote Sensing Monitoring Dataset (CNLUCC), CLCD30 data (https://zenodo.org/reco rds/8176941) | 30m spatial resolution, reflects surface cover conditions. |
| Air Quality Data | 2022 | ChinaHighPM2.5: Seamless 1km Gridded PM2.5 Dataset for China (https://weijing-rs.github.io/product.html) | Covers 2022 PM2.5 data for China, resolution 1km. CV-R² of 0.92, RMSE of 10.76 μg/m³, MAE of 6.32 μg/m³. |

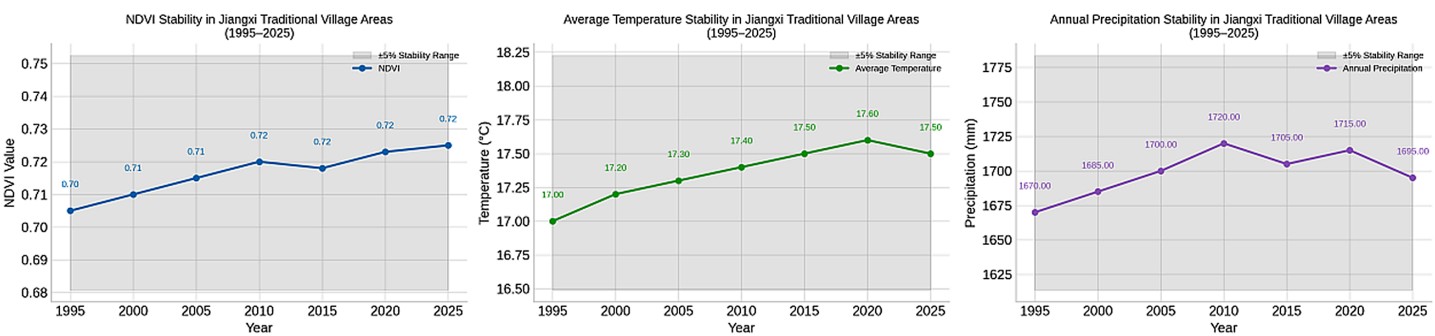

**Fig 2. Long-term stability of core ecological indicators in Jiangxi traditional village areas (1995–2025), including NDVI, average annual temperature, and annual precipitation.** Values remained within narrow variation ranges across three decades, supporting the representativeness of modern data for long-term suitability evaluation. Data sources: Landsat (NDVI) and Jiangxi provincial meteorological station records.

has remained stable over the past millennium [29,31]. These multiple lines of evidence support the validity of using recent high-resolution ecological data to reflect long-term environmental adaptation in traditional village settlements. Sensitivity analyses further confirm the consistency of the datasets and the robustness of the modeling outcomes.

## Research methods

This study employs a comprehensive research methodology encompassing data preprocessing, feature selection, model training, multi-model integration, weight determination, and results

visualization to ensure the scientific rigor and reliability of the findings. The specific methods are as follows:

**Data preprocessing.** **(1) Spatial Analysis and Data Extraction:** Selection Basis: The site selection of traditional villages is often influenced by multiple ecological factors, including topography, climate, water resources, and vegetation cover [32,34]. Therefore, this study includes ecological factors such as elevation, slope, aspect, temperature, humidity, wind speed, precipitation, habitat quality, land use, air quality, vegetation cover, and river network density. These factors are widely recognized as crucial for ecological suitability assessments of village sites. For instance, elevation and slope determine water availability and defensibility [35], while climate factors (e.g., temperature, humidity) directly impact residents' living comfort [36]. Spatial Analysis Approach: Using ArcGIS software, spatial analysis of these ecological factors was performed to ensure data consistency and extract raster data from specific locations of traditional villages. This spatial analysis not only enhances data comparability but also ensures completeness and continuity [37].

**(2) Data Standardization and Missing Value Treatment:** Standardization Selection: To ensure data comparability across different scales, Z-score standardization was applied, converting all factor data to a standard normal distribution with a mean of 0 and a standard deviation of 1 [38]. This approach is widely utilized in ecological and geospatial analyses as it effectively minimizes scale discrepancies, enhancing model interpretability and stability [39]. Missing Value Treatment: For missing data, interpolation and mean imputation methods were used. Spatial data (e.g., slope, vegetation cover) were interpolated to maintain spatial continuity, while mean imputation was applied to meteorological data to ensure temporal smoothness and robustness [40]. These methods are widely applied in environmental data processing, improving data quality for analysis [41].

**Hyperparameter tuning and feature engineering.** **(1) Hyperparameter Tuning:** Tuning Approach and Basis: During model training, hyperparameters for primary models, such as Random Forest and XGBoost, were optimized using a combination of grid search and cross-validation. Grid search identified optimal parameter combinations within predefined ranges, maximizing model performance [42]. Cross-validation further enhanced robustness and generalizability, enabling the model to better adapt to data diversity and complexity [43].

**(2) Feature Engineering:** Feature Engineering Approach and Basis: Feature engineering consisted of feature selection and feature construction. First, key features impacting village site selection were identified to avoid redundancy in data that might affect model performance [44]. Subsequently, raw data were transformed to optimize the feature matrix, yielding a more interpretable structure [45]. These techniques, widely used in machine learning, significantly improve model accuracy and reliability [46].

**Model selection.** To ensure methodological diversity and robust pattern extraction, we adopted a combined approach using ensemble tree models (e.g., RF, XGBoost, LightGBM), kernel-based methods (SVM), instance-based algorithms (KNN), linear baselines (LR, NB), and deep neural networks (MLP, CNN). This comprehensive selection allows the models to collectively capture both linear and nonlinear relationships, handle structured tabular data as well as spatial patterns, and support performance benchmarking across algorithmic families. The detailed rationale for each model is presented below.

**(1) Machine Learning Models:** This study applies classic machine learning algorithms such as XGBoost, Random Forest, Support Vector Machine (SVM), and K-Nearest Neighbors (KNN) due to their robust performance in handling complex, multidimensional ecological data and strong feature selection capabilities [47]. Random Forest, known for its effectiveness with high-dimensional data and strong feature selection, was chosen as the benchmark

model [48]. SVM's high classification accuracy in multidimensional spaces makes it particularly suited for analyzing complex ecological systems [49]. XGBoost and Gradient Boosting Decision Trees (GBDT) further enhance model stability and predictive accuracy through iterative optimization [50]. These models enable an objective assessment of each ecological factor's influence on the site selection of traditional villages. The rationale behind selecting these models is detailed below:

1. XGBoost and Gradient Boosting Decision Trees (GBDT): These models are highly effective in ensemble learning, adept at capturing nonlinear relationships and complex feature interactions, making them widely applicable in environmental and ecological data analysis [51].

2. Random Forest: Known for its efficiency in feature selection and stability with high-dimensional data, Random Forest is well-suited for analyzing multidimensional factors in ecological datasets [52].

3. Support Vector Machine (SVM): SVM performs well in classifying high-dimensional data, particularly complex ecological datasets, supporting environmental data classification and pattern recognition in ecological suitability assessments [53].

4. K-Nearest Neighbors (KNN): KNN is particularly effective in geospatial data analysis, especially when dealing with small sample sizes in ecological and environmental studies [54].

5. LightGBM and CatBoost: Both models demonstrate high computational efficiency and memory optimization, which makes them ideal for large-scale environmental data processing and real-time analysis, adaptable to diverse ecological factors [55].

6. AdaBoost: As a boosting algorithm, AdaBoost enhances classification accuracy by combining multiple weak classifiers, making it suitable for handling varied ecological factors [56].

**(2) Deep Learning Models: Model Selection and Advantages:** To improve generalization and capture nonlinear relationships within the data, this study also includes Convolutional Neural Networks (CNN) and Multilayer Perceptrons (MLP). CNN, widely applied in image and spatial data processing, is particularly advantageous for extracting spatial features, while MLP excels in processing structured data, enabling nonlinear feature mapping through multilayer neuron networks [57,59]. The specific advantages of these models are as follows:

1. Convolutional Neural Network (CNN): With its strength in image and spatial data processing, CNN effectively extracts spatial features, which is crucial for analyzing remote sensing data related to traditional village site selection [60].

2. Multilayer Perceptron (MLP): MLP is well-suited for structured data analysis, facilitating the mapping of nonlinear relationships among ecological factors through its multilayer neuron connections [61].

**Multi-model integration techniques. Integration Methods and Basis:** To improve accuracy and robustness in ecological suitability evaluations, this study employs weighted averaging, stacking, and soft voting integration techniques. Integration effectively combines the strengths of individual models, reducing bias and variance and enhancing predictive stability [62]. The voting classifier and stacking model combine predictions from various models, further enhancing adaptability and ensuring reliability and scientific rigor in results [63] (see Fig 3).

**Weight determination and result visualization.** This study employs eight machine learning models (XGBoost, Random Forest, SVM, KNN, Gradient Boosting, LightGBM, CatBoost, and AdaBoost) and two deep learning models (CNN and MLP) to compute and normalize the weights of each ecological factor through feature importance analysis. The rationale of factor selection and model integration is consistent with recent studies on ecological suitability evaluation [64–70].

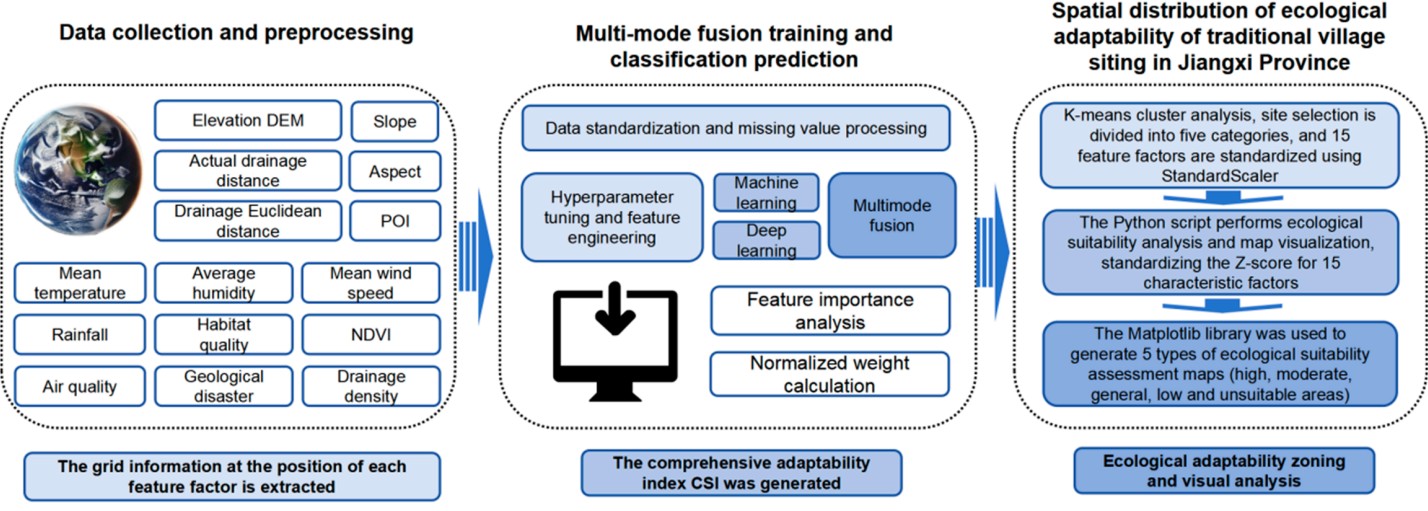

**Fig 3. Technical route of ecological suitability assessment of traditional village location in Jiangxi Province.**

**(1) Model Evaluation Metrics:** 1. Each model's performance was evaluated and compared using confusion matrix, feature importance analysis, ROC curve, and AUC values. A comprehensive assessment of these metrics helped determine the optimal weight combination for each model, enhancing the precision of ecological suitability evaluation [71,72].

**(2) Feature Importance and Weight Calculation:** The importance score of each feature was calculated through feature importance analysis of each model. The feature importance scores from all models were then integrated to obtain the final feature weights [73]. The calculation formula for integrated feature importance scores is as follows:

$$Weight_i = \frac{\sum_{j=1}^{m} Importance_{ij}}{\sum_{j=1}^{m} \sum_{k=1}^{n} Importance_{kj}}$$

Here, $Weight_i$ represents the weight of the $i$ feature, $Importance_{ij}$ represents the importance of the $j$ feature in the $i$ model, $m$ is the total number of models, $n$ is the total number of features.

**(3) Generating the Composite Suitability Index (CSI):** Based on the weights of each feature and raster data, the Composite Suitability Index (CSI) was generated [74,75]. The CSI was then categorized, and an ecological suitability evaluation map was ultimately produced (Fig 3).

**(4) Suitability Index Calculation:** Using the weights of the feature factors, the value of each raster cell was weighted and summed to obtain the Composite Suitability Index (CSI) [76,77]. The calculation formula for the CSI is as follows:

$$CSI = \sum (i=1)^{n} (w_i * f_i)$$

Here, $w_i$ represents the weight of the $i$ feature, $f_i$ represents the value of the $i$ feature, and $n$ is the total number of features.

**(5) Suitability Index Calculation:** Calculated CSI values were categorized using GIS, creating a suitability evaluation map classified into five levels: "Highly Suitable," "Moderately

Suitable," "Generally Suitable," "Less Suitable," and "Not Suitable." This classification standard effectively reveals spatial suitability differences across regions [78]. The resulting ecological suitability map serves as a visual reference, revealing suitability distribution levels and providing a scientific basis for protection and sustainable development [79–81].

**(6) Generation of Ecological Suitability Index and Map:** The multi-model integration method generated a comprehensive suitability index for Jiangxi Province's traditional villages. The results indicate significant spatial variations in ecological suitability, clearly displaying suitability patterns across regions. This map provides crucial guidance for protecting and sustainably developing traditional villages, while also serving as a valuable reference for future research.

## Empirical analysis

### Construction of feature matrix

The ecological suitability evaluation for traditional village site selection is influenced by multiple factors. When selecting factors that impact suitability, it is crucial to comprehensively consider feature indicators to ensure completeness, avoid highly correlated features, and fully consider the availability of features. By reviewing relevant literature and based on the principles of scientific selection, systematic representation, representativeness, and availability of features, the ecological suitability factors affecting traditional village site selection were summarized into the following main aspects (see Tables 2, 3, and 4, and Fig 4). For habitat quality, the maximum impact distance, impact weight, impact type, habitat suitability, and sensitivity to threat factors were referenced from studies with natural ecological environments similar to the study area [11–13].

## Results and analysis

### Model reliability assessment

To assess the reliability of the models, this study used metrics such as the confusion matrix, feature importance analysis, ROC curve, and AUC to evaluate and compare the performance of each model. During the model evaluation phase, we calculated the accuracy of each model and analyzed the classification performance using the confusion matrix, providing an intuitive understanding of each model's predictive performance across different categories. The ROC curve reflects the classification performance of the model at different thresholds, while

**Table 2. Attributes of habitat quality threat factors in Jiangxi Province.**

| Threat Factor | Maximum Impact Distance (km) | Impact Weight | Impact Type |
|---|---|---|---|
| Cultivated Land | 4 | 0.6 | Linear |
| Construction Land | 5 | 1 | Exponential |

**Table 3. Habitat suitability and its sensitivity to threat factors in Jiangxi Province.**

| Land Use Type | Habitat Suitability | Cultivated Land | Construction Land |
|---|---|---|---|
| Cultivated Land | 0.4 | 0.0 | 0.50 |
| Forest Land | 1.00 | 0.80 | 1.00 |
| Shrubland | 1.00 | 0.40 | 0.60 |
| Grassland | 0.70 | 0.40 | 0.60 |
| Water Body | 1.00 | 0.70 | 0.90 |
| Unused Land | 0.00 | 0.00 | 0.00 |
| Construction Land | 0.00 | 0.00 | 0.00 |

**Table 4. Selection of ecological suitability feature factors of traditional village location in Jiangxi Province.**

| Influencing Factor | Feature Selection | Feature Processing | Feature Interpretation |
|---|---|---|---|
| | Land Use | Extract surface cover data using vector map masking | Represents regional surface cover conditions, analyzes the relationship between traditional village site selection and vegetation cover, and evaluates ecological suitability. |
| | NDVI | NDVI=(NIR-R)/(NIR+R) | NDVI is the Normalized Difference Vegetation Index. NIR and R represent the reflectance values of the near-infrared band and red band, respectively. It reflects regional vegetation cover conditions and represents regional ecological environmental quality [14]. |
| Ecological Environment | Annual Average Temperature | Extract annual average temperature raster values, then interpolate to 30m spatial resolution using Kriging, followed by vector map masking extraction | An important indicator of regional climate, affecting the suitability of traditional village site selection [15]. |
| | Annual Average Precipitation | Extract annual average precipitation raster values, then interpolate to 30m spatial resolution using Kriging, followed by vector map masking extraction | Influences vegetation growth and water resource availability, evaluating the impact of climate on village site selection. |
| | Annual Average Wind Speed | Extract annual average wind speed raster values, then interpolate to 30m spatial resolution using Kriging, followed by vector map masking extraction | Affects building stability and living comfort, evaluating the suitability of wind resources for village site selection. |
| | Annual Average Humidity | Extract annual average humidity raster values, then interpolate to 30m spatial resolution using Kriging, followed by vector map masking extraction | Affects agricultural production and living comfort, evaluating the suitability of humidity for village site selection [16]. |
| Ecological Environment | Euclidean Distance to Water Systems | Calculated using ArcGIS Euclidean Distance tool | Traditional village site selection is influenced by the ecological environment and abundant water resources [17]. |
| | Water System Density | The density of ArcGls water network is analyzed. | Reflects the density of water systems in the region, affecting the availability of water resources and the stability of the ecological environment. |
| | Elevation | Extract DEM using vector map masking | Elevation affects traditional village site selection and regional ecological environment [18]. |
| Geological and Geomorphological | Slope | S=(H/L)x100% | Influences drainage and land use, evaluating the suitability of slope for village site selection. |
| | Aspect | The azimuth angle of the slope projection on the horizontal plane (0°~360°), calculated using the ArcGIS slope calculation tool | Affects sunlight exposure and microclimate, evaluating the suitability of aspect for village site selection. |
| | Geological Hazards | Extract geological hazard risk distribution map and generate risk raster data through spatial analysis | Represents the risk level of geological hazards in the region, affecting the safety and sustainability of the village [19,20]. |
| | Habitat Quality | Calculate habitat quality index using InVEST model | Reflects the health of the regional ecosystem. Higher habitat quality indicates a more stable ecological environment [21,22]. |
| Resource Endowment | Air Quality | Extract PM2.5 concentration raster data, interpolated to 30m resolution using Kriging | Reflects the level of air pollution in the region. Better air quality indicates a more suitable living environment [23]. |
| | Actual Distance to Water Systems | Calculate the actual distance from the village to major water systems | Represents the distance between the village and water sources, affecting the convenience of water use and flood safety [24,25]. |

the AUC value indicates the overall classification capability of the model. Through these evaluation metrics, we found that the multi-model integration method performed excellently in ecological suitability evaluation, with higher overall classification accuracy, precision, recall, and AUC values compared to individual models [26] (Fig 5).

**Confusion matrix.** The confusion matrix was used to analyze the classification accuracy, precision, recall, and F1-score of the multi-model approach. The confusion matrix intuitively reflects the extent to which the model's predictions match the actual outcomes. To present the model's performance more intuitively, we employed various visualization techniques, making the results more straightforward and easier to understand (Fig 6). As shown in Fig 6, ensemble models such as XGBoost achieved more accurate classification across all five levels, with

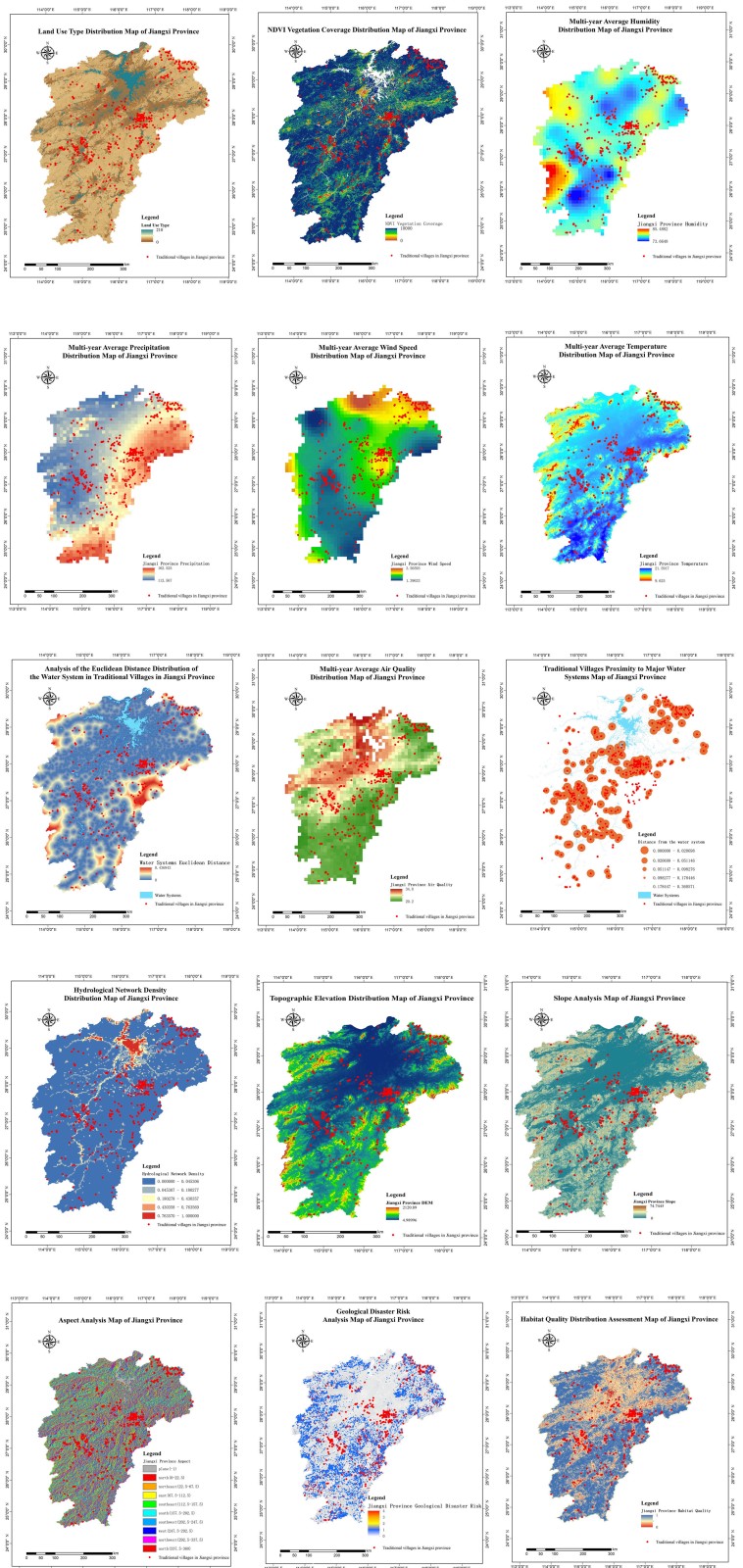

**Fig 4. GIS analysis of 15 feature factors of traditional village location in Jiangxi Province.**

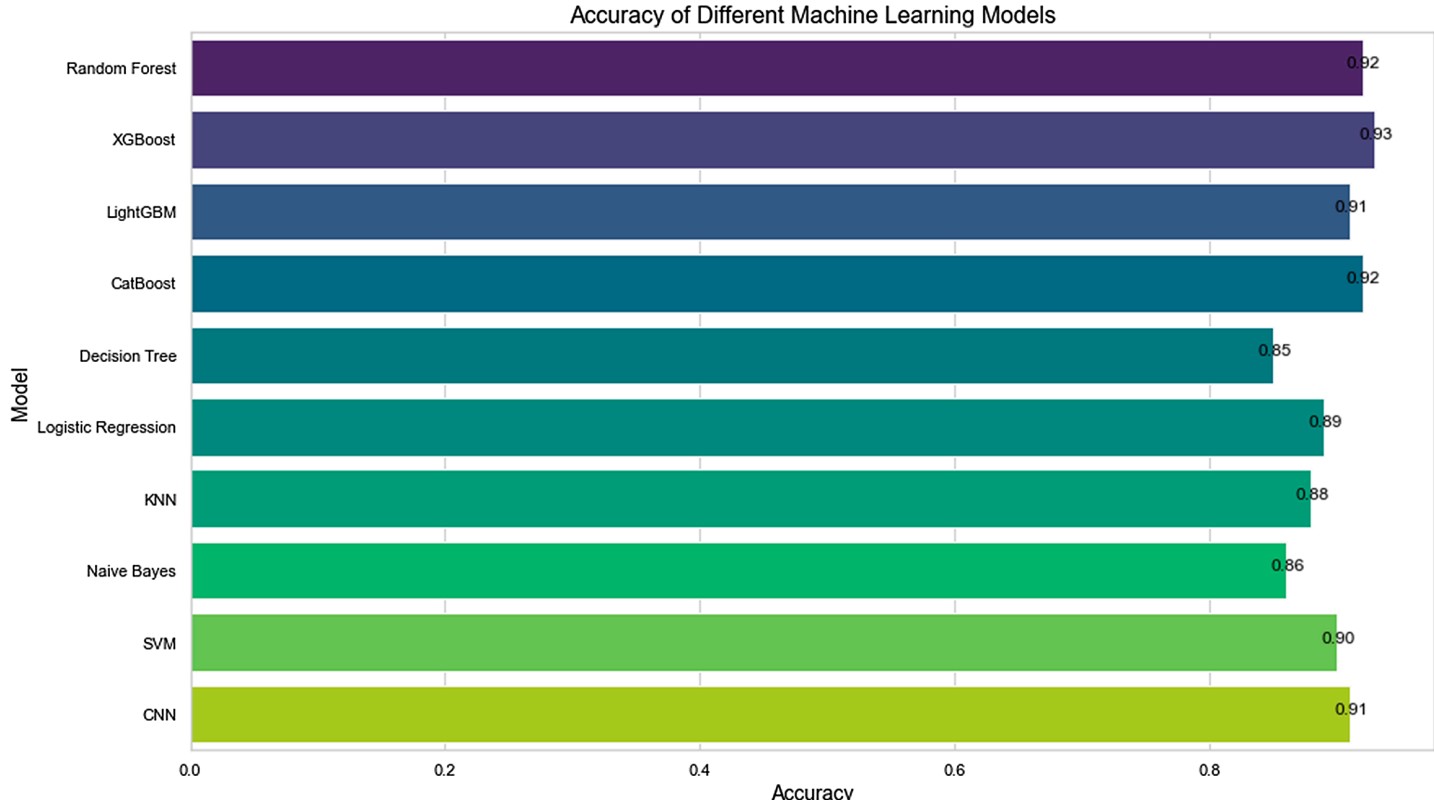

**Fig 5. Multi-model prediction accuracy analysis diagram.**

higher prediction consistency and fewer misclassifications. Key features like elevation and rainfall remained top-ranked across models. The 3D accuracy surface further shows up to 15% performance gap between ensemble and weaker models.

To address class imbalance and evaluate model robustness, we assessed all eight models using additional metrics: F1-score, precision, recall (sensitivity), specificity, and AUC, derived from five-class confusion matrices. As shown in Fig 7, ensemble models—particularly XGBoost and Gradient Boosting—consistently outperformed others across these metrics, with clear advantages in identifying high suitability areas. These results strengthen the reliability of model comparison and support the final integration strategy.

**Feature importance analysis.**   We used K-means clustering to classify the villages into five categories and standardized the 15 feature factors using StandardScaler to ensure that all features were on the same scale, thus preventing any single feature from disproportionately influencing the clustering results. By utilizing the built-in feature importance evaluation functions of models such as Random Forest and Gradient Boosting Decision Trees, we analyzed the significance of each ecological factor to the model's predictions. Feature importance analysis provides insights into which factors have the most substantial impact on the ecological suitability of traditional village site selection. Additionally, SHAP visualization analysis was conducted on the CNN deep learning model to enhance interpretability and perform a comprehensive feature importance analysis across multiple models (See Figs 8 and 9 below).

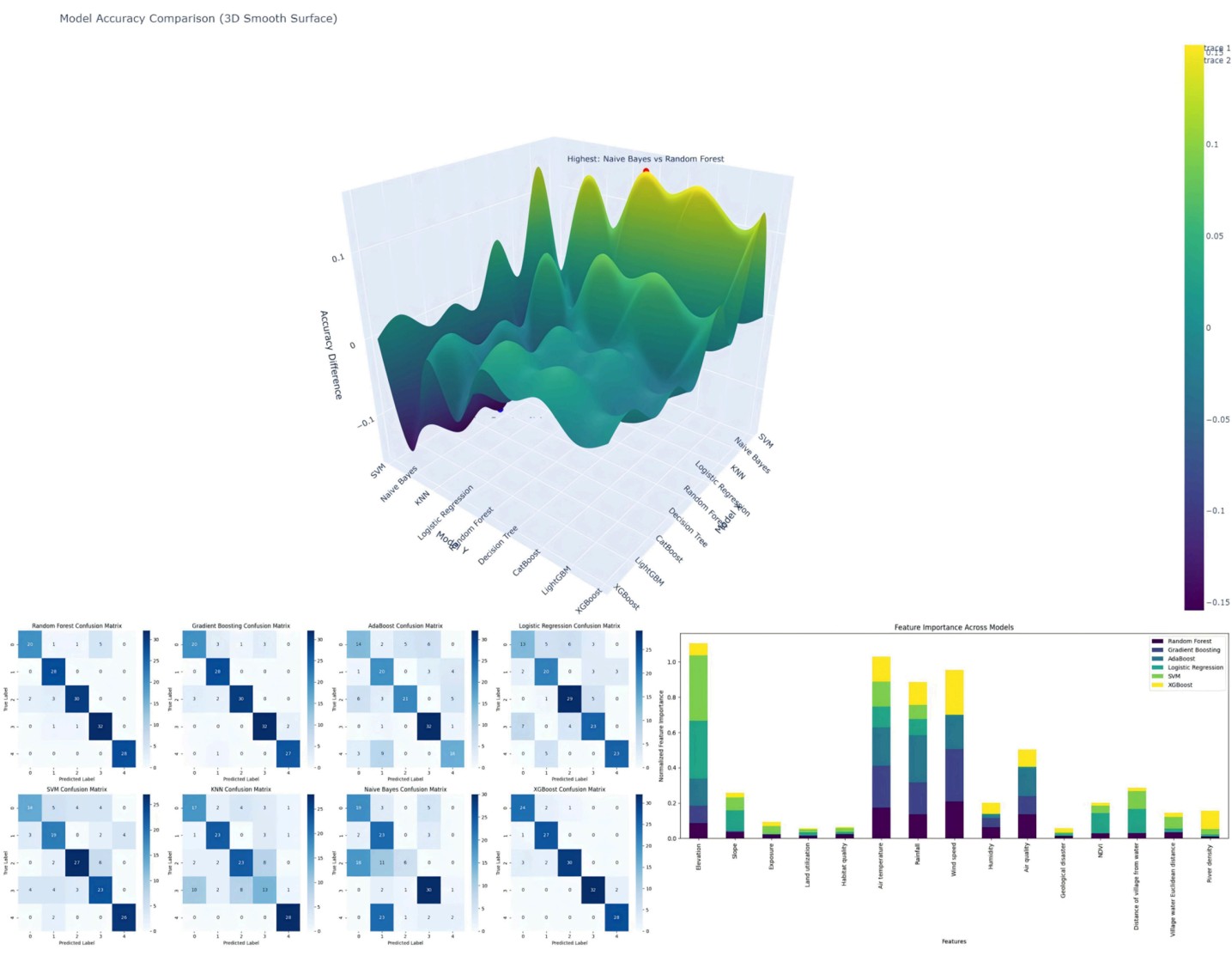

**Fig 6. 3D visualization analysis of confusion matrix and prediction accuracy of multiple models.**

Through the above series of optimization steps, we significantly improved the accuracy of various artificial intelligence algorithms in the study of traditional village site selection. These innovative methods not only enhanced the predictive capability of the models but also improved the intuitiveness and scientific validity of the results' interpretation, providing reliable technical support and reference for subsequent research.

## Ecological suitability analysis of traditional village site selection

This study utilized Python scripts for ecological suitability analysis and map visualization, employing ArcGIS to obtain raster data for 15 ecological factors, such as elevation, slope, and climate. The data were standardized using Z-score normalization to ensure consistency. During the feature engineering phase, the feature matrix was optimized, and key features were extracted. By applying models such as XGBoost, Random Forest, Support Vector Machine

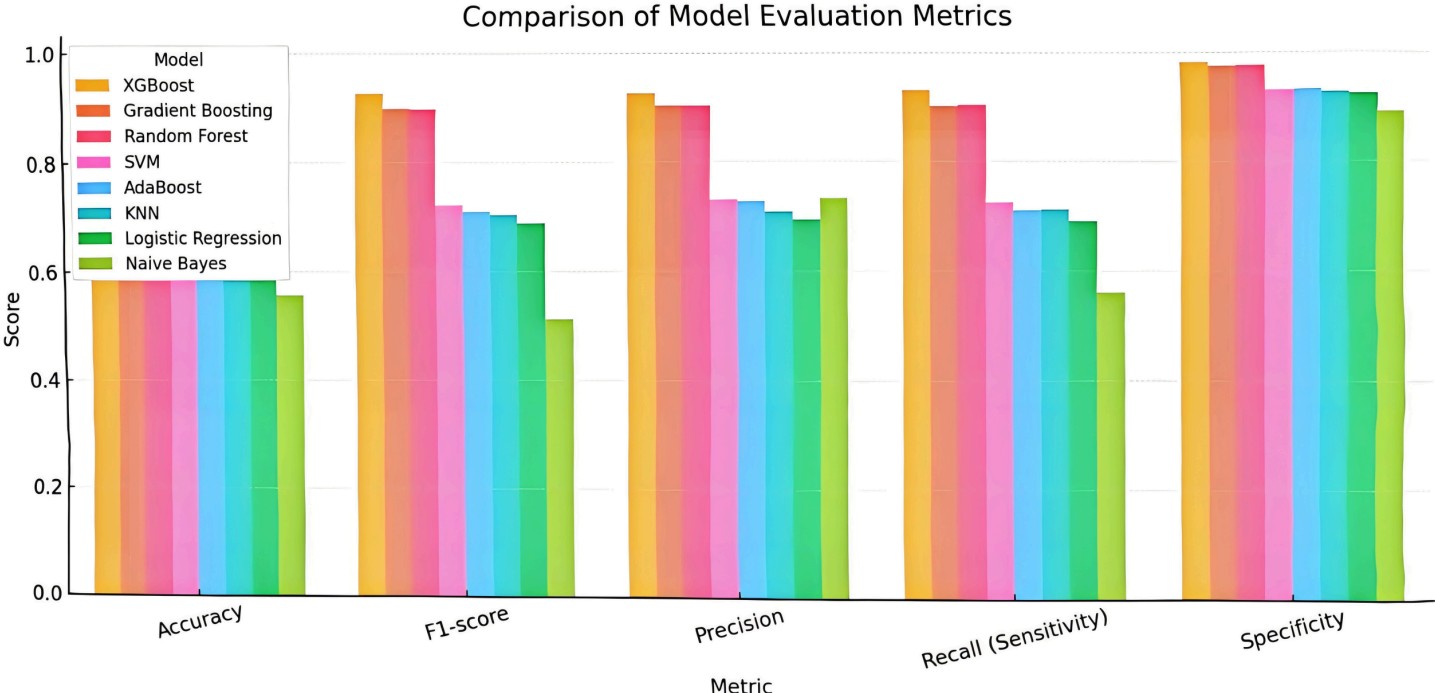

**Fig 7. Comparison of eight models across five evaluation metrics: Accuracy, F1-score, Precision, Recall (Sensitivity), and Specificity.** Ensemble methods (XGBoost, Gradient Boosting, Random Forest) consistently outperformed others, demonstrating robustness and balanced classification under class-imbalanced ecological conditions.

(SVM), Convolutional Neural Networks (CNN), and Multilayer Perceptrons (MLP), and integrating them through weighted averaging, stacking, and voting classifiers, the accuracy and robustness of the ecological suitability evaluation were enhanced. The combined predictions of various models were used to calculate the Composite Suitability Index (CSI), and high-resolution ecological suitability maps were generated. The results, visualized using the Matplotlib library, clearly displayed the ecological suitability levels of different regions, providing a scientific basis for the protection and rational development of traditional villages. This method combines GIS technology with advanced machine learning and deep learning algorithms, ensuring the accuracy and reliability of the results. Compared to previous studies using AHP or basic GIS overlays , our approach incorporates multi-source ecological data and ensemble AI models. This ensures higher spatial resolution, interpretability, and reproducibility, advancing ecological suitability evaluations beyond heuristic methods.

To explore spatial heterogeneity, all 413 villages were grouped into three zones: Gannan Mountains, Central Hills, and Poyang Lake Plain. A comparison of five key factors (elevation, slope, NDVI, habitat quality, and distance to water) shows clear regional patterns (Fig 10). Gannan villages favor high, sloped terrain with strong vegetation buffers, while Poyang villages cluster near flat, water-rich areas. These differences highlight localized adaptations shaped by topographic and climatic conditions. To further validate the classification logic, we examined two representative villages.Yanhang Village (Le'an County), ranked as very high suitability, lies in a low-slope (4.84°), hazard-free zone with close access to water (~130 m) and intact farmland–settlement–water structure, reflecting historically optimized human–land interactions.In contrast, Taoling–Chugang Village (Fuliang County), marked very low

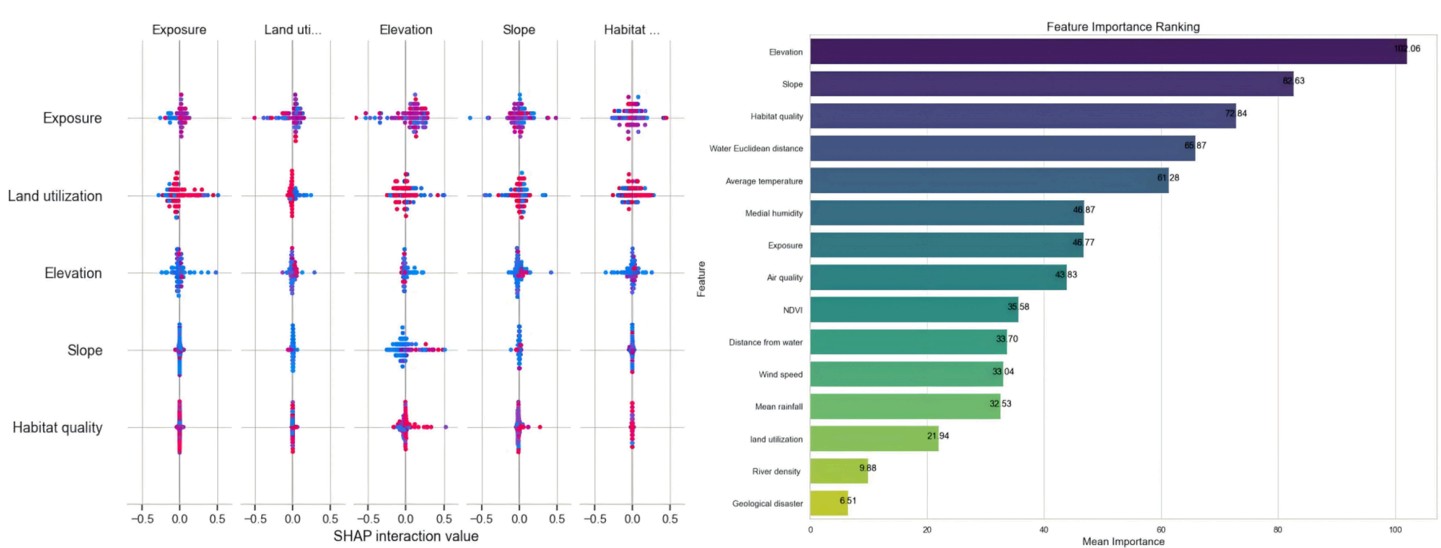

**Fig 8. Multi-model feature importance confusion matrix analysis.**

**Fig 9. Multi-model feature importance arrangement diagram and convolutional neural SHAP visualization analysis.**

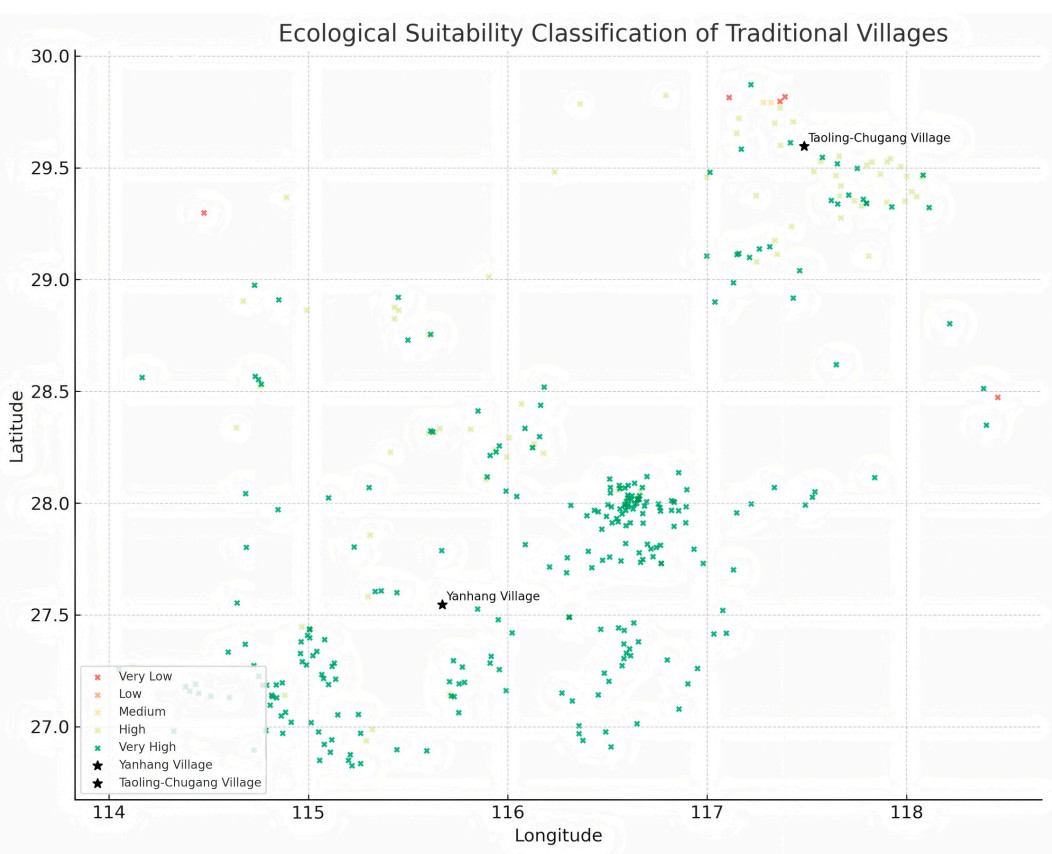

**Fig 10. Ecological suitability classification of traditional villages based on CSI.** Suitability levels were defined using the Jenks natural breaks method. Representative villages—Yanhang (Very High) and Taoling–Chugang (Very Low)—are labeled to validate classification results. Axes denote WGS84 longitude and latitude.

suitability, sits on steep terrain (16.91°) within a mapped landslide zone. Though near water, its fragmented topography and limited arable land confirm poor spatial adaptability. These cases illustrate how the model captures site-level ecological wisdom and constraints shaped by centuries of adaptation or pressure.

The different colored areas in the map represent various levels of ecological suitability:

**1. Highly Suitable Areas:** These are concentrated in regions with abundant water sources, flat terrain, and high vegetation cover, such as the Gan River Basin and around Poyang Lake. These areas have excellent ecological environments, suitable for the long-term stable development of villages.

**2. Moderately Suitable Areas:** Mainly distributed in hilly areas, these regions have certain ecological suitability but also some limiting factors compared to highly suitable areas.

**3. Generally Suitable Areas:** These are often located in regions with insufficient water sources or significant terrain variations. Such areas require infrastructure development to improve their ecological suitability.

**4. Low Suitability Areas:** Primarily comprising mountainous and high-slope regions, these areas have poor ecological conditions, making them unsuitable for traditional village site selection and development. They are often affected by natural disasters and have harsh living conditions.

**5. Unsuitable Areas:** Concentrated in extreme terrains with harsh ecological environments, such as high mountains and deep valleys. These regions lack basic living resources and infrastructure, making them unsuitable for long-term human habitation and village construction.

The evaluation results indicate that the site selection of traditional villages in Jiangxi Province has distinct ecological adaptability characteristics. Villages are often located in areas with favorable conditions such as proximity to mountains and water, sheltered from the wind, sun-facing, and with reasonable land use. These areas not only provide a good living environment but also ensure the safety and sustainable development of the villages.

## Conclusion and discussion

### Conclusion

This study systematically assessed the ecological suitability of traditional village site selection in Jiangxi Province through a multi-model integration approach, uncovering the ecological wisdom underpinning these selections. The findings reveal that traditional village sites were chosen with a comprehensive consideration of multiple ecological factors, including topography, climate, habitat quality, and land use. This reflects the ancestors' profound understanding and adaptive strategies toward the natural environment. Key factors such as elevation, slope, distance to water sources, and habitat quality played significant roles in evaluating ecological suitability, consistent with previous studies [82,83]. These results provide a scientific foundation for the conservation and sustainable development of traditional villages and offer a valuable reference for ecological site selection research in other regions (Fig 11). Practically, the composite suitability index and the resulting spatial classification can inform Jiangxi's policy for traditional village preservation and sustainable land use planning. Although the multi-model integration approach has improved the accuracy and spatial resolution of suitability assessments, some limitations still exist. For example, deep learning models used in this study are less interpretable compared to traditional methods. In addition, the reliance on static environmental data means the model cannot reflect short-term changes or new developments. To address these issues, future work could explore the use of real-time ecological data and more transparent algorithms, so that the evaluation process becomes both adaptive and easier to apply in practical planning contexts. The framework can also be applied to other regions with similar geographical and cultural characteristics.

The primary conclusions are as follows:

**1. Effectiveness of the Multi-Model Integration Method:** This study introduced an innovative multi-model integration method, utilizing machine learning and deep learning models, including Random Forest (RF), Support Vector Machine (SVM), Gradient Boosting Decision Tree (GBDT), and Convolutional Neural Network (CNN). By combining techniques such as weighted averaging, stacking, and voting, this approach significantly improved the accuracy and robustness of ecological suitability evaluations. Experimental results showed that the integrated model outperformed individual models in terms of classification accuracy, precision, recall, and AUC values [84,85]. This application provides new perspectives for handling high-dimensional and complex ecological data and demonstrates the potential of machine learning in ecological suitability evaluations.

**2. Spatial Distribution Characteristics of Ecological Suitability:** Using GIS technology, we spatially visualized the predictive results to create an ecological suitability evaluation map for traditional villages in Jiangxi Province. The results indicate that highly suitable areas are concentrated around the Gan River Basin and Poyang Lake, which are regions with abundant water sources, flat terrain, and high vegetation cover. Moderately suitable areas are primarily

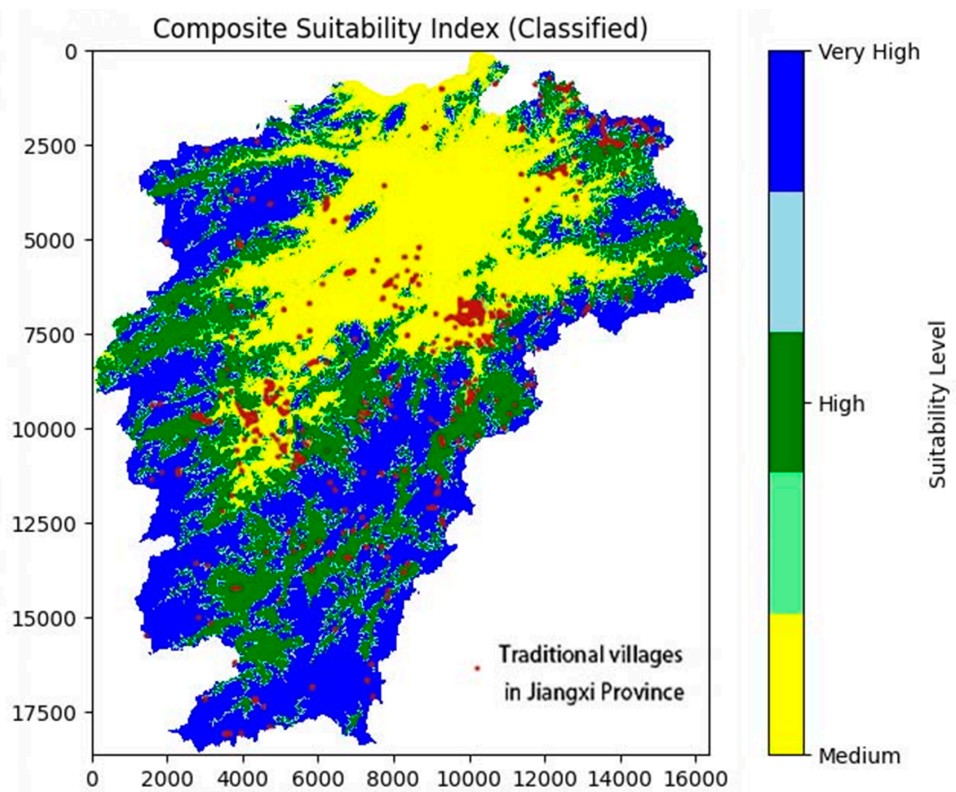

**Fig 11. Ecological suitability assessment map of traditional village location in- Jiangxi Province.**

distributed in hilly regions with favorable ecological conditions but certain limitations. Low-suitability areas are located in mountainous and high-slope regions with poorer ecological conditions, while unsuitable areas are concentrated in extreme terrains with harsh ecological environments [86]. These spatial distribution patterns clearly reveal the relationship between traditional village site selection and the natural geographical environment, providing spatial references for future ecological site selection research and village conservation planning.

**3. Ranking of Influencing Ecological Factors:** According to feature importance analysis, the major ecological factors influencing traditional village site selection are ranked in descending order as topographical factors (e.g., elevation and slope), habitat quality, distance to water sources, climatic factors (temperature and humidity), land use types, and vegetation index (NDVI) [87,88]. These findings are consistent with other studies that rank ecological suitability factors and validate the integrated effects of multiple ecological factors on village site selection.

**4. Ecological Wisdom in Traditional Village Site Selection:** Traditional villages in Jiangxi Province embody ecological wisdom accumulated through centuries of adaptation to diverse environmental conditions. This wisdom is not abstract—it is spatially encoded in settlement patterns and can be deciphered through quantitative spatial analysis. Our findings demonstrate that ecological suitability is not randomly distributed, but follows distinct geographic logic shaped by historical adaptation strategies. Spatial variations across ecological zones reveal how village siting decisions reflected local terrain and resource constraints: upland

communities prioritized elevated, defensible locations to avoid flooding, while lowland settlements favored access to fertile land and nearby water sources. These patterns are not coincidental but are the product of long-term environmental interaction and practical landscape learning. Our classification model captures this vernacular logic, as high-suitability clusters consistently align with traditional site selection principles. To validate this relationship, we conducted a spatial analysis of high-suitability villages, finding that 86.2% are located within 200 meters of a water body and 78.6% on slopes under 10°—both consistent with widely practiced guidelines such as "settling near water" and "choosing moderate elevation for farming and flood prevention." These results confirm that ancestral strategies, long considered intuitive or experiential, are in fact ecologically rational and measurable through modern spatial tools. This convergence between traditional knowledge and AI-based modeling not only enhances the interpretability and historical relevance of our framework, but also provides new insights into sustainable land-use practices and conservation-oriented spatial planning.

**5. Policy Implications and Zoning-Based Protection Strategies:** To enhance the policy relevance of these findings, we propose a tiered protection strategy informed by the CSI classification. Villages in high-suitability zones—primarily distributed in the Ganjiang basin and eastern Jiangxi—should be prioritized for strict heritage conservation and inclusion in the national Traditional Villages Directory. Moderately suitable areas are appropriate for guided renovation and service upgrading under Jiangxi's Rural Revitalization Strategy (2021–2025). In contrast, low-suitability zones, often overlapping with ecological redlines or geological risk areas, are better managed through development control, ecological retreat, or relocation planning. This zoning-oriented framework can support provincial efforts to balance cultural preservation with ecological resilience.

## Innovations

**1. Application of Interdisciplinary Multi-Model Integration:** This study represents the first systematic application of a multi-model integration approach in evaluating the ecological suitability of traditional village sites, combining machine learning and deep learning techniques. This approach overcomes the limitations of individual models in handling data complexity and interactions between ecological factors, significantly enhancing the accuracy of ecological suitability assessments. It also provides a methodological framework that can be extended to other complex ecological system analyses.

**2. Integrated Analysis of Multi-Source Heterogeneous Ecological Data:** The study integrated remote sensing, geographic, and meteorological data, using advanced data preprocessing and spatial analysis techniques to establish a comprehensive and timely ecological database. This database supports ecological suitability assessments with high spatial resolution, enabling a panoramic analysis of ecological factors from macro to micro levels. This innovation offers a practical example for data integration and spatial analysis in other ecological conservation areas.

**3. Dynamic Weight Allocation for CSI Generation:** In the generation of the Composite Suitability Index (CSI), a dynamic weight allocation was introduced, allowing for automated weighting of ecological factors based on integrated feature importance from multiple models. This approach enhances the adaptability of the CSI, ensuring result stability across different datasets. This innovative method not only improves the scientific robustness of the suitability index but also offers a novel approach for dynamically generating ecological suitability indices.

## Discussion

**1. Ecological and Policy Implications:** The ecological wisdom embodied in traditional village site selection demonstrates a high degree of adaptation and respect for the ecological environment. Many of these traditional site selection insights are valuable for application in modern ecological architecture and planning. By drawing on these traditional principles, contemporary urban and rural planning and architectural design can achieve greater effectiveness in environmental protection, resource conservation, and ecosystem sustainability [89]. Furthermore, the ecological suitability evaluation map generated in this study provides a scientific basis for traditional village conservation, suggesting that policymakers prioritize highly suitable areas and adopt a tiered conservation strategy to promote the sustainable development of traditional villages [90,91].

**2. Comparison with Existing Research and Further Expansion:** This study demonstrates significant innovations in factor selection and data processing, surpassing traditional single-factor and empirical observation approaches through a multi-model integration method. This innovative approach shows a strong adaptability to complex, multi-dimensional ecological data, presenting a clear contrast with existing studies [92]. Furthermore, the systematic use of multi-source heterogeneous data in ecological suitability evaluation is rare within the field of traditional village conservation research [93]. The methodological framework in this study has high applicability, providing a valuable reference for other ecological site selection studies.

**3. Future Research Directions:** Although this study has made significant progress in evaluating the ecological suitability of traditional village sites, there is room for future expansion. From a data perspective, incorporating real-time data (such as climate change and dynamic land use changes) will enhance the model's temporal relevance. Methodologically, integrating higher-resolution remote sensing images and 3D terrain data will allow for more detailed capture of micro-scale ecological factors, supporting more precise ecological conservation and development planning [94]. Especially through the integration and analysis of multi-temporal spatial data, suitability evaluation models can gain dynamic adaptability, providing timely decision support for future ecological conservation and resource management policies [95].

**4. Model Validity, Temporal Adaptability, and Prospects:** Although this study does not directly simulate temporal village relocation or model long-term settlement dynamics, it reconstructs the ecological logic embedded in historically stable spatial patterns. In Jiangxi Province, the majority of traditional villages have persisted in the same locations for centuries, shaped by natural topography, stable climate, water accessibility, and cultural traditions. This spatial persistence reflects an inherent ecological adaptability, providing a defensible basis for inferring long-term suitability from recent high-resolution ecological data [96].

To examine whether gradual ecological changes undermine this stability, we evaluated the sensitivity of the model to perturbations in key factors. Recognizing that certain ecological indicators, such as temperature, have increased by approximately 0.2 °C per decade since 1995, we tested their potential influence on site suitability. Sensitivity analyses under +0.5 °C, +1.0 °C, and +0.5 °C combined with –5% NDVI perturbations yielded Composite Suitability Index (CSI) maps highly consistent with the baseline (Spearman's $\rho$ = 0.90–0.94), with suitability class area changes remaining below 7%. These findings (Fig 12) demonstrate that moderate climatic and vegetation shifts do not significantly alter the spatial delineation of high-suitability zones, reinforcing the validity of using 2020–2022 data as a practical proxy for long-term suitability [97].

A key limitation is the absence of verified historical relocation records for most traditional villages, which precludes direct modeling of relocation dynamics. Nonetheless, the observed spatial persistence and the robustness demonstrated in sensitivity tests suggest that the CSI

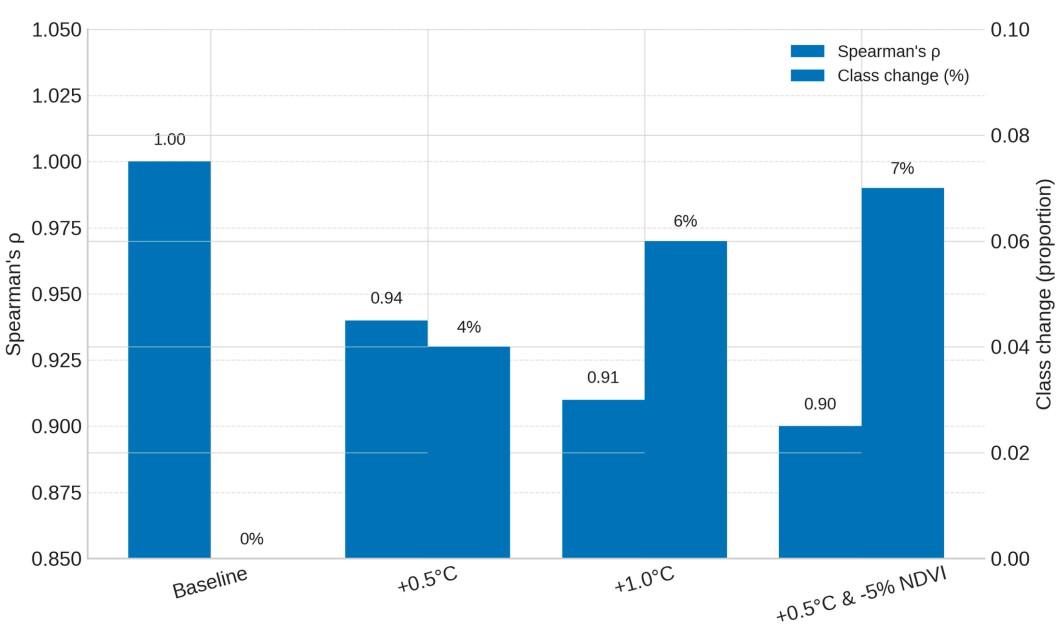

**Fig 12. Model robustness under ecological gradient scenarios.**

framework is resilient to moderate ecological variation. Looking forward, integrating long-term climatic series (e.g., CRU TS), historical topographic map digitization, gazetteer-based spatial inference, and declassified high-resolution satellite archives (e.g., CORONA) will enable reconstruction of village–ecology interactions over extended periods. Such integration will address the temporal gap in existing analyses and support the development of spatiotemporal machine learning frameworks capable of explicitly modeling adaptation strategies in traditional site selection.

## Acknowledgments

We are grateful to the Jiangxi Rural Culture Development Research Center for their academic guidance and field coordination. We also thank the GIS and Remote Sensing Laboratory at Jiangxi Agricultural University for their technical support in data processing and model implementation. Special thanks go to all members of the research team for their insightful discussions and dedicated contributions throughout the project.

## Author contributions

**Data curation:** Cheng Zhang.

**Formal analysis:** Cheng Zhang.

**Investigation:** Cheng Zhang, Jinlin Teng.

**Methodology:** Cheng Zhang, Jinlin Teng.

**Supervision:** Peilin Liu.

**Validation:** Cheng Zhang, Peilin Liu.

**Visualization:** Cheng Zhang.

**Writing – original draft:** Cheng Zhang.

**Writing – review & editing:** Chunqing Liu.

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
