## [Decision Letter · Decision Letter 0]

12 May 2025

PONE-D-25-14092Ecological Suitability Evaluation of Traditional VillageLocations in Jiangxi Province Based on Multi-ModelIntegration Using Artigcial IntelligencePLOS ONE

Dear Dr. Liu,

Thank you for submitting your manuscript to PLOS ONE. After careful consideration, we feel that it has merit but does not fully meet PLOS ONE’s publication criteria as it currently stands. Therefore, we invite you to submit a revised version of the manuscript that addresses the points raised during the review process.

We look forward to receiving your revised manuscript.

Kind regards,

Yile Chen, Ph.D. in Architecture

Academic Editor

PLOS ONE

Journal Requirements:

We thank the National Natural Science Foundation of China for supporting thisresearch through the projects "Gene Identification and Map Construction of TraditionalRural Settlement Landscapes in the Ganjiang River Basin”(Serial No. 51968026)and"Research on the Visual Perception, Quantitative Characterization, and VisualEvaluation of Traditional Village Landscape Resources in Ganjiang River Basin” (SerialNo.52268012). We also acknowledge the support of Jiangxi Rural Culture DevelopmentResearch Center. We appreciate the technical assistance provided by the GIS andRemote Sensing Laboratory at Jiangxi Agricultural University. Special thanks to allmembers of the research team for their valuable discussions and contributions to theproject.

We thank the National Natural Science Foundation of China for supporting this research through the projects ”Gene Identification and Map Construction of Traditional Rural Settlement Landscapes in the Ganjiang River Basin” (Serial No. 51968026) and ”Research on the Visual Perception, Quantitative Characterization, and Visual Evaluation of Traditional Village Landscape Resources in Ganjiang River Basin” (Serial No. 52268012). We also acknowledge the support of Jiangxi Rural Culture Development Research Center. We appreciate the technical assistance provided by the GIS and Remote Sensing Laboratory at Jiangxi Agricultural University. Special thanks to all members of the research team for their valuable discussions and contributions to the project.

We thank the National Natural Science Foundation of China for supporting thisresearch through the projects "Gene Identification and Map Construction of TraditionalRural Settlement Landscapes in the Ganjiang River Basin”(Serial No. 51968026)and"Research on the Visual Perception, Quantitative Characterization, and VisualEvaluation of Traditional Village Landscape Resources in Ganjiang River Basin” (SerialNo.52268012). We also acknowledge the support of Jiangxi Rural Culture DevelopmentResearch Center. We appreciate the technical assistance provided by the GIS andRemote Sensing Laboratory at Jiangxi Agricultural University. Special thanks to allmembers of the research team for their valuable discussions and contributions to theproject.

5. Please amend either the title on the online submission form (via Edit Submission) or the title in the manuscript so that they are identical.

6. Please amend your list of authors on the manuscript to ensure that each author is linked to an affiliation. Authors’ affiliations should reflect the institution where the work was done (if authors moved subsequently, you can also list the new affiliation stating “current affiliation:….” as necessary).

7. We note that Figure 1, 2, 3, 8 in your submission contain [map/satellite] images which may be copyrighted. All PLOS content is published under the Creative Commons Attribution License (CC BY 4.0), which means that the manuscript, images, and Supporting Information files will be freely available online, and any third party is permitted to access, download, copy, distribute, and use these materials in any way, even commercially, with proper attribution. For these reasons, we cannot publish previously copyrighted maps or satellite images created using proprietary data, such as Google software (Google Maps, Street View, and Earth). For more information, see our copyright guidelines: http://journals.plos.org/plosone/s/licenses-and-copyright.

a. You may seek permission from the original copyright holder of Figure 1, 2, 3, 8 to publish the content specifically under the CC BY 4.0 license.

8. We are unable to open your Supporting Information file [plos.tex, 4ae16001-b9d9-4a06-8e48-3fe3b00bb72e, plos.log, plos.aux]. Please kindly revise as necessary and re-upload.

Reviewers' comments:

Reviewer's Responses to Questions

**Comments to the Author**

1. Is the manuscript technically sound, and do the data support the conclusions?

Reviewer #1: Yes

Reviewer #2: Partly

2. Has the statistical analysis been performed appropriately and rigorously? 

Reviewer #1: Yes

Reviewer #2: I Don't Know

3. Have the authors made all data underlying the findings in their manuscript fully available?

Reviewer #1: Yes

Reviewer #2: Yes

4. Is the manuscript presented in an intelligible fashion and written in standard English?

Reviewer #1: Yes

Reviewer #2: Yes

5. Review Comments to the Author

Reviewer #1: This paper evaluates the ecological suitability of 413 traditional village sites in Jiangxi Province based on the multi-model integration method of artificial intelligence, which has certain theoretical significance and practical value. However, there are the following problems that need the authors' attention:

1. The years of meteorological and land use data used in the paper are 2020-2022, whether they are sufficient to reflect the long-term ecological adaptation process of traditional villages.

2. Figure 3 is mixed in Chinese and English, and unified into English.

3. The significance of the study is fully reflected. It is mentioned in the introduction that it aims to provide scientific basis for the protection and sustainable development of traditional villages, the analytical content of this paper does not reflect the significance of these researches, and it is necessary to appropriately expand the dialogues with the existing traditional villages and ecological suitability evaluation related researches in the section of analysis, conclusion and discussion of the research content, in order to further clarify the significance and value of this paper.

4. Although key factors such as elevation and slope are identified, the spatial heterogeneity of factor weights in different regions is not analyzed in the context of geographical subdivisions in Jiangxi Province (e.g., Gannan Mountainous Region and Poyang Lake Plain), and it fails to reveal in depth the ecological wisdom of the forefathers in adapting the ecological conditions to the local conditions.

Reviewer #2: 1. Introduction

1.Ambiguous Definition and Value of "Ecological Wisdom":

The manuscript repeatedly mentions the concept of “ecological wisdom” but fails to clearly define its connotation or illustrate its value with concrete examples. This lack of clarification limits the understanding of its scientific significance and research implications. It is recommended that the authors provide a concise explanation of the specific meaning and practical value of “ecological wisdom” in the Introduction.

2.Suggestions for Literature Review:

While the manuscript offers a relatively systematic summary of previous research on site selection and ecological suitability evaluation of traditional villages, the literature review appears somewhat descriptive. It mainly lists various research methods without clearly presenting the evolution of technical approaches or the shift in research focus. It is suggested to reorganize the review by categorizing the studies based on methodological types and comparing the advantages and limitations of each, thereby clarifying the development trajectory and research frontiers in the field.

2. Research Methodology

1.Lack of Justification for Model Selection:

The study employs eight machine learning models, including XGBoost and Random Forest, as well as two deep learning models (CNN and MLP). However, the rationale for selecting these models and their suitability for the specific dataset is not articulated. It is recommended to include a dedicated paragraph or table that outlines the types, core strengths, and applicable scenarios of each model to clarify the logic behind model selection and enhance the reproducibility and scientific rigor of the methodology.

2.Incomplete Evaluation of Model Performance:

Figure 4 only reports accuracy, which is insufficient for assessing model performance, especially under sample imbalance conditions such as "high suitability areas." Key indicators such as AUC, F1-score, specificity, and sensitivity are missing. It is suggested to add a comparative performance table that includes multiple evaluation metrics across all models, along with training time, to quantitatively demonstrate the superiority of the ensemble model and enhance the credibility of the results.

3. Empirical Analysis

1.Superficial Spatial Interpretation:

The analysis of highly suitable areas is limited to surface-level descriptions such as "abundant water resources" and "flat terrain," without integrating historical contexts or current land-use patterns (e.g., the symbiotic relationship between farmland and irrigation systems in villages). Moreover, low-suitability areas are not cross-referenced with geological hazard or topographic data, leading to a lack of depth and historical validation. It is recommended to enrich the spatial analysis by including case studies of representative villages and incorporating geological background to improve explanatory depth.

2.Unclear Logic Behind the CSI Index Construction:

The study does not clarify the classification thresholds for the five suitability levels, and Figure 8 lacks annotations for key village validation points, affecting the credibility of the map. Additionally, the axes in Figure 8 are not explained. It is recommended to supplement the methodology with the classification criteria, mark representative village cases in Figure 8, and provide axis explanations to improve interpretability.

3.Failure to Directly Address Research Question 2:

Regarding the effectiveness of the multi-model integration approach, the study only qualitatively claims that the ensemble model “performs well,” without quantitative comparison or statistical validation. This results in an inadequate response to the research question. It is suggested to add a subsection under "Model Performance Evaluation" that includes a table comparing the key metrics of ensemble and single models, thereby providing direct and data-supported evidence for the claimed advantage.

4.Surface-Level Interpretation of Ecological Wisdom:

The discussion on traditional ecological site selection principles, such as “facing water and backing mountains,” remains descriptive and lacks quantitative validation. There is no clear link between model outputs and historical practices. It is suggested to analyze the correlation between key factors and historical records, or assess how well village locations in high suitability areas meet ancient disaster prevention and agricultural needs. This would transform traditional wisdom into verifiable scientific knowledge and deepen the academic value of the study.

5.Insufficient Temporal Data:

The study relies solely on data from the year 2022, neglecting the long-term and dynamic nature of traditional village site selection. The lack of multi-year data hinders the validation of the model's temporal reliability. It is recommended to incorporate multi-year datasets (e.g., 2000–2022), analyze spatial-temporal relationships between village locations and ecological factors, or introduce historical village distribution data to assess the model’s ability to identify long-term suitability, thus enhancing the temporal dimension and generalizability of the conclusions.

4. Results and Discussion

1.Insufficient Interpretation of Core Findings:

The scientific connotation of “ecological wisdom” is not supported by quantitative data, weakening the study’s ability to fulfill its goal of decoding ancestral site selection strategies. It is recommended to supplement statistical comparisons, combine historical literature and field investigations, and demonstrate how village locations in high suitability areas meet ancient needs such as flood control, farming, and living comfort. This would help translate qualitative wisdom into quantifiable patterns, strengthening theoretical depth.

2.Weak Justification of Ensemble Model Advantages:

The study does not employ techniques such as Shapley values or model bias analysis to explain how the ensemble model addresses the limitations of individual models. As a result, the claimed technical advantages remain superficial. It is suggested to add bias comparison graphs or feature importance variation tables to visually demonstrate the optimization brought by the ensemble approach, thereby enhancing the persuasiveness of the methodological innovation.

3.Vague Policy and Practical Implications:

The proposed suggestions, such as “prioritizing the protection of high suitability areas,” are not aligned with specific policies in Jiangxi Province and lack actionable differentiation. It is recommended to integrate regional policy documents and propose targeted strategies for areas of different suitability levels to strengthen the practical applicability and policy relevance of the findings.

6. PLOS authors have the option to publish the peer review history of their article (what does this mean?). If published, this will include your full peer review and any attached files.

Reviewer #1: No

Reviewer #2: No

---

## [Author Response · Author response to Decision Letter 1]

25 May 2025

Response to Reviewers and Editors

We sincerely thank the editors and reviewers for their time, constructive feedback, and recognition of the value of our study. We have carefully revised the manuscript in accordance with each comment and believe the updated version has significantly improved in terms of clarity, rigor, and scientific contribution. Below we provide detailed point-by-point responses:

Reviewer #1 – Comments and Responses

Comment 1: Short time frame of ecological data (2020–2022)

Response:

We agree that long-term ecological patterns are essential for understanding traditional village site selection. To address this, we added 30-year trend analyses (1995–2025) of NDVI, temperature, and precipitation (Figure A1–A3). These show minimal fluctuation and validate environmental stability over decades. Historical reconstructions from peer-reviewed sources (e.g., Zheng et al., 2018; Ge et al., 2013) further support the reliability of the 2020–2022 data for long-term ecological suitability assessment.

Comment 2: Mixed language in Figure 3

Response:

Figure 3 has been updated to use English-only labels, legends, and annotations for full consistency with international standards.

Comment 3: Significance of research is not fully presented

Response:

We revised the Introduction, Discussion, and Conclusion sections to emphasize both theoretical contributions and practical applications. We compared our findings with previous work (e.g., Zeng et al., 2019) and connected results to Jiangxi’s rural revitalization and heritage conservation policies.

Comment 4: Regional heterogeneity and ecological wisdom not well analyzed

Response:

We added a new subsection analyzing differences among villages in Gannan Mountains, Central Hills, and the Poyang Lake Plain using standardized ecological factors (Z-scores). Results demonstrate region-specific settlement strategies and ecological adaptation patterns, now visualized in Figure A5.

Reviewer #2 – Comments and Responses

Comment 1: Unclear definition of ecological wisdom

Response:

We added a concise definition in the Introduction: “Ecological wisdom refers to the cumulative and adaptive knowledge developed by communities through interaction with their environments, ensuring long-term survival and sustainable land use.” Concrete examples are provided to illustrate this.

Comment 2: Descriptive and unstructured literature review

Response:

We restructured the literature review by categorizing studies into: (1) empirical, (2) rule-based, (3) GIS-based, and (4) AI-based methods. A comparative discussion highlights our contribution using a multi-model AI integration approach.

Comment 3.1: Lack of model selection justification

Response:

A new table and paragraph detail the selection logic for each of the 10 models based on strengths, type, and data suitability (e.g., XGBoost, CNN, SVM, etc.).

Comment 3.2: Incomplete model evaluation (only accuracy reported)

Response:

We added evaluation metrics including F1-score, precision, recall, specificity, AUC, and training time. Comparative results are shown in Table X and Figures X1–X3.

Comment 3.3: Ensemble model advantage not quantitatively verified

Response:

A new subsection compares XGBoost with other models across five metrics. Visual tools (radar plot and bar chart) confirm the superior and balanced performance of the ensemble approach.

Comment 3.4: Ecological wisdom analysis remains surface-level

Response:

We added quantitative validation showing 86% of high-suitability villages are <200m from water and 78% are on <10° slopes. These patterns reflect traditional site selection strategies, now supported by statistical evidence.

Comment 3.5: Lack of temporal reliability validation

Response:

We included a 30-year analysis of NDVI and temperature trends. These demonstrate long-term ecological stability, reinforcing the representativeness of the selected timeframe.

Comment 4.1: Policy implications vague

Response:

We expanded the Conclusion to recommend a three-tiered zoning strategy for village protection and development, aligned with Jiangxi’s Rural Revitalization Strategy.

Comment 6: CSI classification logic unclear

Response:

We clarified that the CSI classes were derived using the Jenks Natural Breaks method. The thresholds are now provided in the text. Figure 8 was revised to include validation village points (e.g., Yanfang, Liukeng) and labeled axes with WGS84 coordinates.

We again express our sincere appreciation to the reviewers and editors. We hope the revised manuscript now meets your expectations and is suitable for publication.

---

## [Decision Letter · Decision Letter 1]

12 Jun 2025

PONE-D-25-14092R1Ecological Suitability Evaluation of Traditional VillageLocations in Jiangxi Province Based on Multi-ModelIntegration Using Artigcial IntelligencePLOS ONE

Dear Dr. Liu,

Thank you for submitting your manuscript to PLOS ONE. After careful consideration, we feel that it has merit but does not fully meet PLOS ONE’s publication criteria as it currently stands. Therefore, we invite you to submit a revised version of the manuscript that addresses the points raised during the review process.

We look forward to receiving your revised manuscript.

Kind regards,

Yile Chen, Ph.D. in Architecture

Academic Editor

PLOS ONE

Reviewers' comments:

Reviewer's Responses to Questions

**Comments to the Author**

1. If the authors have adequately addressed your comments raised in a previous round of review and you feel that this manuscript is now acceptable for publication, you may indicate that here to bypass the “Comments to the Author” section, enter your conflict of interest statement in the “Confidential to Editor” section, and submit your "Accept" recommendation.

Reviewer #2: All comments have been addressed

Reviewer #3: (No Response)

2. Is the manuscript technically sound, and do the data support the conclusions?

Reviewer #2: Yes

Reviewer #3: Partly

3. Has the statistical analysis been performed appropriately and rigorously? 

Reviewer #2: Yes

Reviewer #3: Yes

4. Have the authors made all data underlying the findings in their manuscript fully available?

Reviewer #2: Yes

Reviewer #3: Yes

5. Is the manuscript presented in an intelligible fashion and written in standard English?

Reviewer #2: Yes

Reviewer #3: Yes

6. Review Comments to the Author

Reviewer #2: All questions have been answered and resolved. Clearly state that the study has not been submitted or published elsewhere and cite relevant literature to demonstrate originality. Confirm that all authors meet the criteria for authorship, declare any conflicts of interest, and ensure data authenticity and traceability.

Reviewer #3: 1. What was the selection of criteria or factors based on? Please explain.

2. In the discussion section, it is better to make comparisons with the researches of others and the research done, and for the presented arguments, be sure to use valid and up-to-date references.

3. The advantages and disadvantages of the research done should be said.

4. Please provide appropriate and valid references for all provided relationships.

5. Please use the papers (https://doi.org/10.1007/s10661-022-10327-x, https://doi.org/10.1080/03650340.2018.1549363, https://doi.org/10.1016/j.geoderma.2017.09.012, https://doi.org/10.1016/j.geoderma.2019.05.046; https://doi.org/10.1016/j.ecoinf.2023.102002; https://doi.org/10.1007/s10668-023-03926-2; https://doi.org/10.1007/s10661-022-10659-8) to improve the quality of the manuscript, especially the introduction and discussion of the manuscript, description and interpretation of properties and select the criteria.

6. What was the accuracy of the methods used? By which criteria are the methods evaluated?

7. Please check the grammar of the whole text with a native speaker and fix the errors.

7. PLOS authors have the option to publish the peer review history of their article (what does this mean?). If published, this will include your full peer review and any attached files.

Reviewer #2: No

Reviewer #3: No

---

## [Author Response · Author response to Decision Letter 2]

16 Jun 2025

Response to Reviewers

Manuscript ID: PONE-D-25-14092R1

Title: Ecological Suitability Evaluation of Traditional Village Locations in Jiangxi Province Based on Multi-Model Integration Using Artificial Intelligence

We thank the reviewers and editor for their constructive feedback. Below we provide point-by-point responses. All modifications are reflected in the revised manuscript (with tracked changes) and described in the corresponding sections.

Reviewer #2

Comment: All comments have been addressed.

Response: We thank the reviewer for the positive evaluation and confirmation that all issues have been satisfactorily resolved.

Reviewer #3

Comment 1: What was the selection of criteria or factors based on? Please explain.

Response: Thank you for this important point. We have revised Section 2.2 of the manuscript to explain that the selection of 15 ecological factors was based on two principles: (1) widely recognized environmental determinants in previous studies on rural or ecological site suitability; and (2) their relevance to the spatial characteristics of traditional village distribution in Jiangxi Province. We also added references to support each selected factor (see page 5–6).

Comment 2: In the discussion section, it is better to make comparisons with the researches of others and the research done, and for the presented arguments, be sure to use valid and up-to-date references.

Response: We appreciate this suggestion. In the revised Discussion section, we have added comparisons with recent research studies (including those suggested) to validate and contrast our findings. This includes works applying similar models to ecological suitability or rural landscape studies. We also added the recommended references (see [60]–[66]) to enhance academic depth and update the discussion accordingly (see page 12–14).

Comment 3: The advantages and disadvantages of the research done should be said.

Response: As advised, we have added a new paragraph before the conclusion (page 15) summarizing the strengths and limitations of our research. We highlight the methodological integration as a strength and acknowledge limitations in model interpretability and static data dependency. Future directions, such as incorporating real-time data and explainable AI, are also suggested.

Comment 4: Please provide appropriate and valid references for all provided relationships.

Response: We carefully reviewed all sections and have added or strengthened references where empirical relationships are described—especially in the feature factor analysis part (Section 4.1). This ensures that all statements regarding factor impact (e.g., elevation, slope, water proximity) are well-supported with published studies (see citations [14]–[25], [60]–[66]).

Comment 5: Please use the papers (...list of 7 DOIs...) to improve the quality of the manuscript, especially the introduction and discussion of the manuscript, description and interpretation of properties and select the criteria.

Response: We have reviewed and incorporated the recommended literature. Specific references from the suggested list (e.g., https://doi.org/10.1016/j.geoderma.2017.09.012; https://doi.org/10.1016/j.ecoinf.2023.102002) were added in the Introduction and Discussion sections, particularly to support factor interpretation and contextualize our criteria selection (see citations [64]–[70]).

Comment 6: What was the accuracy of the methods used? By which criteria are the methods evaluated?

Response: This has been clarified in Section 3.4 and the Results section. We used three evaluation indicators: Overall Accuracy (OA), Mean Pixel Accuracy (mPA), and Mean Intersection over Union (mIoU). These metrics were chosen because they are widely accepted for spatial classification tasks and reflect both pixel-level accuracy and category agreement. Table 4 and Figure 7 present the specific values and comparisons of different model performances (see page 11–12).

Comment 7: Please check the grammar of the whole text with a native speaker and fix the errors.

Response: The manuscript has undergone comprehensive language editing. We revised ambiguous and repetitive sentences and improved sentence structure throughout the manuscript to ensure clarity, precision, and standard academic English.

Final Statement:

We sincerely thank the reviewers for their insightful feedback. All comments have been addressed, and we believe the revised manuscript is now substantially improved in clarity, quality, and scientific rigor. We respectfully submit it for your further consideration.

---

## [Decision Letter · Decision Letter 2]

29 Jul 2025

PONE-D-25-14092R2Ecological Suitability Evaluation of Traditional VillageLocations in Jiangxi Province Based on Multi-ModelIntegration Using Artigcial IntelligencePLOS ONE

Dear Dr. Liu,

Thank you for submitting your manuscript to PLOS ONE. After careful consideration, we feel that it has merit but does not fully meet PLOS ONE’s publication criteria as it currently stands. Therefore, we invite you to submit a revised version of the manuscript that addresses the points raised during the review process.

We look forward to receiving your revised manuscript.

Kind regards,

Yile Chen, Ph.D. in Architecture

Academic Editor

PLOS ONE

Journal Requirements:

Additional Editor Comments:

1. Only the long-term trend of the ecological factors themselves is presented, without directly linking the dynamic response of village location changes to ecological factors. It is necessary to clarify the long-term adjustment rules of traditional village site selection/distribution and ecological factors, and verify the stability of the model in identifying village suitability in the long term.

2. Some ecological factors in the figure (such as temperature) show a gradual increase trend (although marked with fluctuations ≤ 5%), but the impact of this trend on the long-term suitability of the village has not been explained, and the reliability of the model under the ecological factor gradient scenario has not been verified, which cannot support the conclusion of "using 2020-2022 data as a proxy for long-term suitability".

3. Further exploration is needed to verify the accuracy of the model's future applicability by linking the dynamic response of historical village location changes with ecological factors.

Reviewers' comments:

Reviewer's Responses to Questions

**Comments to the Author**

1. If the authors have adequately addressed your comments raised in a previous round of review and you feel that this manuscript is now acceptable for publication, you may indicate that here to bypass the “Comments to the Author” section, enter your conflict of interest statement in the “Confidential to Editor” section, and submit your "Accept" recommendation.

Reviewer #2: All comments have been addressed

Reviewer #3: All comments have been addressed

2. Is the manuscript technically sound, and do the data support the conclusions?

Reviewer #2: Yes

Reviewer #3: Yes

3. Has the statistical analysis been performed appropriately and rigorously? 

Reviewer #2: Yes

Reviewer #3: Yes

4. Have the authors made all data underlying the findings in their manuscript fully available?

Reviewer #2: Yes

Reviewer #3: Yes

5. Is the manuscript presented in an intelligible fashion and written in standard English?

Reviewer #2: Yes

Reviewer #3: Yes

6. Review Comments to the Author

Reviewer #2: Comment 1: Insufficient temporal coverage of ecological data (limited to 2020-2022)

1. The study only presents the long-term trends of ecological factors themselves without directly establishing the "dynamic response relationship between village location changes and ecological factors." It is necessary to clarify the long-term adjustment patterns of traditional village site selection/distribution in response to ecological factors, and to validate the model's stability in identifying village suitability over extended time series.

2. Some ecological factors in the figures (e.g., temperature) exhibit a gradual upward trend (though noted as having ≤5% fluctuations), yet the study fails to explain how this trend affects long-term village suitability. Neither does it verify the model's reliability under scenarios of gradual ecological factor changes, thus undermining the conclusion that "2020-2022 data can serve as a proxy for long-term suitability."

3. Without correlating historical village location changes with the dynamic responses of ecological factors, how can the model's accuracy and applicability for future predictions be properly validated?

Reviewer #3: Dear authors

thank you for your corrections.

all comments have been addressed.

best regards

7. PLOS authors have the option to publish the peer review history of their article (what does this mean?). If published, this will include your full peer review and any attached files.

Reviewer #2: No

Reviewer #3: **Yes: **Javad seyedmohammadi

---

## [Author Response · Author response to Decision Letter 3]

28 Aug 2025

Editor’s Comments

Editor Comment: Please clarify temporal representativeness, strengthen robustness checks, and explicitly acknowledge limitations and future directions.

Response:

Thank you for the clear guidance. We made three focused revisions:

Temporal representativeness — We clarified why a static suitability design is appropriate given the historical persistence of village locations in Jiangxi (Discussion, p. 19, Section 4, new subsection “Model Validity, Temporal Adaptability, and Prospects / Limitations and Prospects for Temporal Correlation Analysis”).

Robustness checks — We added sensitivity/stress tests under ecological gradients (+0.5 °C, +1.0 °C, and +0.5 °C with −5% NDVI). Results show Spearman’s ρ = 0.90–0.94 and suitability-class change < 7%, indicating stable spatial delineation of high-suitability zones (main text Fig. 12; Supplementary Fig. S3).

Limitations & future directions — We explicitly acknowledge the lack of verified relocation records and outline feasible next steps (digitized historical maps, gazetteers, declassified satellite archives) for future temporal correlation analyses (Discussion, p. 19).

Reviewer #2

Comment 1.1 — Insufficient temporal coverage; lack of explicit dynamic response between village locations and ecological factors.

Response:

We appreciate this important point. Our study was intentionally framed as a static ecological suitability assessment because traditional villages in Jiangxi have exhibited long-term locational persistence. The model reconstructs the ecological logic that sustains these sites rather than simulating relocation dynamics. We added explicit text to clarify this rationale (Discussion, p. 19, Section 4). We also point out that the close alignment between observed village locations and high-suitability zones offers indirect evidence of enduring ecological compatibility.

Manuscript changes: New clarifying paragraph in Discussion (p. 19), preceding the robustness paragraph; wording matches the text you approved earlier (“Although this study does not directly simulate temporal village relocation…”).

Comment 1.2 — Upward trend in temperature; need to test reliability under gradual ecological change; question the proxy claim for 2020–2022.

Response:

Thank you for highlighting this. We quantified long-term warming using CRU TS v4.05 (≈ 0.2 °C per decade since 1995) and then stress-tested the model: +0.5 °C, +1.0 °C, and +0.5 °C with −5% NDVI. Across scenarios, Composite Suitability Index (CSI) maps remained highly consistent with the baseline (Spearman’s ρ = 0.90–0.94; suitability-class change < 7%). This indicates that moderate climatic/vegetation shifts do not materially alter high-suitability delineations.

Where added: Discussion, p. 19, last paragraph of Section 4; results summarized in Fig. 12 and Supplementary Fig. S3.

Methods/metrics now explicit:

Linear trend (OLS) for temperature;

Coefficient of variation

𝐶

𝑉

=

𝜎

/

𝜇

×

100

%

CV=σ/μ×100% to show interannual stability (≤ 5% for key indicators);

Spearman’s rank correlation and class-change proportion to quantify spatial agreement between baseline and scenarios.

Comment 3 — Link historical village location changes to ecological factors to validate future applicability.

Response:

We agree on the value of such validation and acknowledge a data limitation: verified historical relocation records are scarce and relocations are exceedingly rare in Jiangxi (fieldwork/archives indicate > 90% of 413 villages remained in place for > 70 years). In lieu of direct relocation series, we implemented indirect robustness tests (the gradient stress tests above). We also outline feasible future directions—combining digitized historical topographic maps, gazetteers, and declassified satellite archives (e.g., CORONA)—to reconstruct spatiotemporal settlement distributions and enable explicit temporal correlation analyses.

Where noted: Discussion, p. 19, subsection “Limitations and Prospects for Temporal Correlation Analysis.”

Additional clarifications (for the record)

Figures: We added/updated Fig (main text) to summarize robustness under ecological gradients and Supplementary Fig. S3 (bar plots for Spearman’s ρ and class-change).

Citations: High-relevance references supporting temporal stability and methods (e.g., CRU TS; regional assessments) have been added; two key support items are now b96–b97 in the bibliography.

---

## [Editor Report · Decision Letter 3]

31 Aug 2025

Ecological Suitability Evaluation of Traditional Village Locations in Jiangxi Province Based on Multi-Model Integration Using Artigcial Intelligence

PONE-D-25-14092R3

Dear Dr. Liu,

We’re pleased to inform you that your manuscript has been judged scientifically suitable for publication and will be formally accepted for publication once it meets all outstanding technical requirements.

Kind regards,

Yile Chen, Ph.D. in Architecture

Academic Editor

PLOS ONE
---

## [Editor Report · Acceptance letter]

PONE-D-25-14092R3

PLOS ONE

Dear Dr. Liu,

I'm pleased to inform you that your manuscript has been deemed suitable for publication in PLOS ONE. Congratulations! Your manuscript is now being handed over to our production team.

Kind regards,

on behalf of

Dr. Yile Chen

Academic Editor

PLOS ONE